# Integrated Optimization of Structure and Control for Fast Steering Mirrors

**DOI:** 10.3390/mi15030298

**Published:** 2024-02-22

**Authors:** Zijie Chen, Qianwen Duan, Luyao Zhang, Yi Tan, Yao Mao, Ge Ren

**Affiliations:** 1National Key Laboratory of Optical Field Manipulation Science and Technology, Chinese Academy of Sciences, Chengdu 610209, China; czijie99@163.com (Z.C.); duanqianwen16@mails.ucas.ac.cn (Q.D.); 202311060903@std.uestc.edu.cn (L.Z.); maoyao@ioe.ac.cn (Y.M.); renge@ioe.ac.cn (G.R.); 2Key Laboratory of Optical Engineering, Chinese Academy of Sciences, Chengdu 610209, China; 3Institute of Optics and Electronics, Chinese Academy of Sciences, Chengdu 610209, China; 4University of Chinese Academy of Sciences, Beijing 100049, China

**Keywords:** fast steering mirror, integrated optimization of structure and control (IOSC), co-simulation, non-dominated sorting genetic algorithm (NSGA)

## Abstract

This study concerns the problem of integrated optimization of structure and control based on a fast steering mirror, aiming to achieve simultaneous optimization of the mechanical structure and control system. The traditional research and development path of the fast steering mirror involves a lengthy process from the initial design to the final physical manufacture. In the prior process, it was necessary to produce physical prototypes for repeated debugging and iterative optimization to achieve design requirements, but this approach consumes a significant amount of time and cost. To expedite this process and reduce unnecessary experimental costs, this study proposes an integrated optimization of structure and control (IOSC) method. With the use of IOSC, it is possible to achieve simultaneous optimization of structure and control. Specifically, the use of non-dominated sorting genetic algorithm II (NSGA-II) obtains globally optimal controller parameters and mechanical structure parameters under certain performance indices. This achieves an effective balance between the resonance frequency generated by the system and the working bandwidth, providing a high-precision reference for the research and development of fast steering mirrors.

## 1. Introduction

The electro-optical tracking system is broadly applied in various fields, such as wireless power transmission, laser communication [1,2], quantum communication [3], and adaptive optics [4]. The fast steering mirror (FSM), which is one of the mechanisms in the electro-optical tracking system, adjusts the propagation direction of the beam between the receiver and the target by controlling the rapid deflection of the FSM. It plays a crucial role in precisely steering the beam, offering preponderance like fast response, high tracking accuracy, and a high working bandwidth [5,6].

Generally, the mechanical structure and control performance of the FSM system limit the working bandwidth [7]. To increase the working bandwidth, it is necessary to design suitable mechanical structures and controllers. Some researchers have proposed methods to increase the working bandwidth of the FSM by optimizing the mechanical structure. For example, D. J. Kluk et al. [8] designed a novel structure with a smaller mass and a more compact structure, increasing the natural frequency of the system and achieving a higher closed-loop bandwidth. W. Zhang et al. [9] used the NSGA-II algorithm to perform multi-objective optimization of the natural frequencies of a two-axis FSM, increasing the system’s working bandwidth through mechanical structure optimization. Others tried to increase the working bandwidth by designing different controllers. Wu [10] discussed a PID controller to suppress resonance in FSM systems. J. Zhong et al. [11] offset resonant peaks in the FSM system through system identification and zero-pole cancellation methods, extending the system bandwidth with the introduction of a feedforward compensator composed of a zero-phase error tracking controller and a zero-phase filter. R. Xiao et al. [12] proposed a bandwidth control strategy combining the incomplete differential PID algorithm and feedforward compensation, significantly increasing the system bandwidth while reducing phase delay. Cao et al. [13] used the hysteresis operator based on the BP neural network to model the FSM hysteresis, effectively described the hysteresis nonlinearity of FSM, and compensated the hysteresis.

However, the studies above are only applicable to the serial design of FSM, where debugging is carried out based on physical prototypes. If the ultimate bandwidth performance fails to meet design requirements, the entire mechanism may need to be redesigned or redone through iterative optimization, leading to additional time and cost. Some researchers have proposed an electro-mechanical integration simulation method based on FSM, completing the design of mechanical and control systems in the design stage. This accelerates the iterative optimization process and provides guidance for the ultimate manufacture. Y. Lu et al. [14] proposed the integrated mechanical control simulation (IMCS) method, predicting the ultimate control performance of the system through simulation. Y. Zhang et al. [15] introduced the electromagnetic model of a voice coil motor into the co-simulation of FSM, which improved the completeness of the simulation. However, these methods belong to a serial design in simulation, where each simulation is only applicable to one model. When the simulation results fail to meet the expected design indices, redesign is still necessary.

To further accelerate the research and development process of FSM, this paper proposes an integrated optimization of structure-control (IOSC) method based on a one-degree-of-freedom (1-DOF) FSM. It should be clarified that FSM systems are mainly divided into 1-DOF systems and 2-DOF systems. In a 1-DOF system, the FSM can only rotate around one axis, while a 2-DOF FSM can rotate around two axes. However, both systems are fundamentally the same in principle. Although 2-DOF FSM systems have broader applications, 1-DOF FSM systems provide lower modeling complexity and manufacturing costs, making them sufficient for validating the method proposed in this paper. Compared with the existing literature [14,15], the method proposed in this paper can simultaneously predict the control performance of the FSM under different structures. Additionally, the method can be applied to higher DOF FSM systems and other mechatronics equipment, which is more universal. The following is the contribution of this study.

(i)To ameliorate the optimization efficiency, the FSM structure parameters and controller parameters are combined to optimize simultaneously.(ii)The multi-objective optimization algorithm is a superior solution when considering multiple mutually exclusive indicators among the structure parameters and controller parameters. Therefore, this study attempts to solve the parameter tuning problem by using a multi-objective optimization algorithm and constructs a non-dominated sorting genetic algorithm II(NSGA-II). By considering the indices with first-order resonant frequency, second-order resonant frequency, and working bandwidth of the system, the final values of the structure parameters and controller parameters can be obtained.(iii)By using this method, the working bandwidth of the FSM is increased while suppressing the first-order resonance of the system.

The remainder of this study is organized below. Section 2 introduces the design of a 1-DOF FSM, providing the working principle of the system and explaining the relationship between structure and bandwidth, with a detailed analysis of the stiffness of the flexible support. Section 3 introduces a method of co-simulation of structure and control for the 1-DOF FSM using multiple software. Section 4 introduces a method that combines structure parameters and controller parameters, outlining how to simultaneously optimize both structure and controller parameters, including controller design and proof of system stability. Section 5 redesigns the FSM based on the optimization results and verifies the effectiveness of the optimization through simulation. Section 6 gives the conclusion of our study.

## 2. Structure Design, Modeling, and Analysis

In this section, we discussed the operational principles of the FSM system in detail and the corresponding mathematical model is given based on these principles. The relationship between the flexible hinge and the system bandwidth is analyzed. Subsequent structural optimization and controller design are both based on this configuration. More details of the FSM are shown in Table 1.

### 2.1. Structural Design and Working Principle

Figure 1 presents the 3D model of the 1-DOF FSM we designed. The entire FSM primarily consists of a mechanical structure, displacement sensor, actuator, and reflective mirror.

The structure of the 1-DOF FSM and the simplified physical model structure are illustrated in Figure 2. The mirror, driven by the actuator, reflects the beacon light into the target detector. The controller works to minimize the error between the position of the beacon light and the centroid of the target detector.

The system shown in Figure 1 is nonlinear and faces nonlinear factors such as friction effects, drive saturation, and unmodeled high-frequency dynamics. In this paper, the FSM is considered a linear system, so we need to make the following assumptions.

Firstly, since the flexible hinge used in our structure is frictionless, the friction effects can be ignored. Secondly, drive saturation is the focus of this paper. We take the stability margin caused by saturation as a performance index to adjust parameters. Thirdly, since the operating frequency of the FSM mainly focuses on low to mid frequencies, the unmodeled high-frequency dynamics can be temporarily ignored. Based on the above assumptions, FSM can be regarded as a linear system.

In Figure 2, the following equations are derived using the principles of electrical balance and torque balance [16].
(1)$$\left\{\begin{array}{c} U_{a}=R_{a}I_{a}\left(s\right)+L_{a}sI_{a}\left(s\right)+K_{b}s\theta_{a}\left(s\right),\\ C_{m}I_{a}=\left(J_{L}s^{2}+f_{m}s+K_{m}\right)\theta_{a}\left(s\right), \end{array}\right.$$
where $U_{a}$, $C_{m}$, $R_{a}$, $I_{a}$, $L_{a}$, $f_{m}$, $K_{b}$, and $K_{m}$ represent the motor voltage, torque coefficient, resistance, current, inductor, viscous friction, back electromotive force coefficient, and spring stiffness, respectively. Concurrently, $J_{L}$ and $\theta_{a}$ represent the load inertia and the relative angle of the motor-driven mirror, respectively. The controlled plant is modeled as follows.
(2)$$G\left(s\right)=\frac{\theta_{a}\left(s\right)}{U_{a}\left(s\right)}=\frac{C_{m}}{\left(J_{L}s^{2}+f_{m}s+K_{m}\right)\left(L_{a}s+R_{a}\right)+K_{b}C_{m}s}.$$

In addition, Equation (Equation 2) can also be decomposed into a second-order oscillatory element and an inertial element, which is
(3)$$G\left(s\right)=\frac{k}{\left(\frac{s^{2}}{\omega_{n}^{2}}+\frac{2\xi s}{\omega_{n}}+1\right)\left(Ts+1\right)},$$
where *k* is the gain of the controlled plant, $\omega_{n}$ and ξ are the natural frequency and damping ratio of the second-order oscillation element, respectively. *T* is the parasitic time constant. The inertial element in Equation (Equation 3) is due to the presence of the motor [15]. When there is no motor in the system, the transfer function of the system can be expressed below.
(4)$$G\left(s\right)=\frac{k}{\left(\frac{s^{2}}{\omega_{n}^{2}}+\frac{2\xi s}{\omega_{n}}+1\right)}.$$

As the frequency of the external force is equal to or even highly close to the natural frequency, the system will undergo resonance, meaning that the output amplitude of the system will increase sharply [17].

In general, for FSM systems with low requirements for accuracy and speed, the working bandwidth is usually narrow, far below the resonant frequency of the structure. The influence of resonance is not a significant concern. However, for fast response and high tracking accuracy, it is necessary to have a high working bandwidth, which may lead to the natural frequency of the system being lower than the system bandwidth, resulting in the influence of resonance. To avoid the impact of resonance, the natural frequency of the system structure can be designed to be much higher than the working bandwidth. However, when the moment of inertia of the system is fixed, the higher the natural frequency, the higher the stiffness of the structure. Therefore, in order to achieve a specific working stroke, the output force of the motor will need to be very large. However, the output force of the motor is generally limited. As the output force increases, the volume and power consumption of the motor will increase, and the internal stress of the structure will increase as well. This approach is not advisable.

Considering that in the working process of the FSM, the control system plays a role in the direction driven by the motor, it is possible to use the controller to suppress the resonance peak caused by the lower stiffness in the working direction of the FSM, while only requiring the natural frequency of the structure in the non-working direction to be much higher than the system’s working bandwidth.

To increase the bandwidth, it is necessary to suppress the resonance peak through control methods. At the same time, the FSM structure should be optimized so that the first-order resonance frequency is as low as possible and the higher-order resonance frequency is as high as possible. The relationship between the natural frequency and the structure is as follows:(5)$$f_{i}=\frac{1}{2\pi }\sqrt{\frac{K_{i}}{J_{i}}},$$
where $f_{i}$ represents the frequency of the structure, $K_{i}$ represents the rotational stiffness of the flexible support in direction *i*, and $J_{i}$ represents the moment of inertia of the system in direction *i*.

### 2.2. Flexure Support Design for Bandwidth Consideration

The flexure support structure of our 1-DOF FSM is shown in Figure 3a, which is based on a tangential circular flexible hinge featuring only one rotational degree of freedom. Its force and deformation are illustrated in Figure 3b, where the stiffness of each degree of freedom can be represented by equations from (6) to (9) [18].
(6)$$K_{\alpha z}=\frac{M_{z}}{\alpha_{z}}=\frac{EbR^{2}}{12f_{1}},$$
where $f_{1}$ can be represented as:(7)$$f_{1}=\frac{12s^{4}\left(2s+1\right)}{{\left(4s+1\right)}^{\frac{5}{2}}}arctg\sqrt[{}]{4s+1}+\frac{2s^{3}\left(6s^{2}+4s+1\right)}{{\left(4s+1\right)}^{2}\left(2s+1\right)},$$
(8)$$K_{\alpha x}=\frac{M_{x}}{\alpha_{x}}=\frac{Eb^{3}}{12}/\left[\frac{2(2s+1)}{\sqrt[{}]{4s+1}}arctan\sqrt[{}]{4s+1}-\frac{\pi }{2}\right],$$
(9)$$K_{y}=\frac{F_{y}}{\Delta y}=Eb/\left[\frac{2(2s+1)}{\sqrt[{}]{4s+1}}arctan\sqrt[{}]{4s+1}-\frac{\pi }{2}\right],$$
where s=R/t.

According to the bandwidth requirements, the natural frequencies of the system structure can be designed using Equations from (5) to (9), and the corresponding structural parameters of the flexible hinge can be calculated. These structural parameters include the gap width *b* of the flexible hinge, the radius *R* of the circular cut, the gap thickness *t*, and the Young’s modulus *E* of the material.

## 3. Co-Simulation

For typical electro-mechanical integrated systems, the mechanical system and control system are series-designed, meaning that the physical prototype of the mechanical system is manufactured first before the design of the control system. This approach prevents us from promptly evaluating the quality of the mechanical system and makes it difficult to fully optimize the electro-mechanical integrated system. The aim is to utilize high-precision simulation technology to solve these problems. Therefore, researchers proposed an integrated mechanical control simulation (IMCS) method [14], which involves using co-simulation of MSC.ADAMS, ANSYS, and Matlab/Simulink to achieve this goal.

Figure 1 shows the 3D model of the 1-DOF FSM, and then a dynamic model of the FSM mechanical system is established in Figure 4. It is worth noting that the model in ADAMS software (Version 2019) is, by default, treated as a rigid body model and considered non-deformable. The flexible hinge in actual fast steering mirrors provides degrees of freedom to the system through its own deformation, making it a flexible body. If the pre-built geometric model is directly imported into ADAMS for kinematic and dynamic analysis, it will lead to failure in dynamic simulation and the flexible hinge being treated as a rigid body, which does not reflect the actual situation. In order to transform the hinge into a flexible body, this paper uses the finite element method (FEM) through ANSYS (Version 2019R2) to convert the hinge into a flexible body model, which is also called Modal Neutral Files (MNF). The ADAMS/Flex interface is utilized in the ADAMS software to achieve data exchange between ANSYS and ADAMS, and the rigid body of the hinge is replaced using the MNF file.

The FSM in this study has only one degree of rotational freedom, and it is controlled by a pair of voice coil motors in a push-pull manner. In ADAMS, the operation of the voice coil motors is simulated using a pair of equally sized and oppositely directed forces to control the mirror, as depicted in Figure 4. These forces are defined as the system input, with the deviation angle of the mirror relative to the balanced position being the system output. Furthermore, utilizing the bi-directional control interface provided by ADAMS/Control, the dynamic model is transferred to Simulink, enabling the establishment of a combined mechanical-control simulation within the Simulink environment.

The co-simulation model is given a swept sine signal. The frequency characteristics of the FSM around the Z-axis are obtained through signal processing, as shown in Figure 5.

By fitting the frequency characteristics in Figure 5, the corresponding transfer function expression is obtained in the following equation.
(10)$$G\left(s\right)=\frac{8.463\times 10^{4}}{s^{2}+53.12s+7.818\times 10^{4}}.$$

This is a typical second-order oscillation element, consistent with Equation (Equation 4) in Section 2. From Figure 5, it can be seen that the system produces a resonance peak at 45 Hz, indicating a first-order resonance frequency of 45 Hz. To validate the accuracy of the co-simulation results, we manufactured a physical prototype, as shown in Figure 6a, which is identical to the simulation model. We conducted the same frequency sweep test on the prototype. Through experimentation, the actual open-loop frequency response results were found to be very similar to the co-simulation results, with minimal error. The matching results of the physical prototype and the experiment with the co-simulation are shown in Figure 6b, demonstrating that co-simulation can predict the dynamic performance of physical prototypes.

As the scanning frequency increases, the signal excitation will cause resonance in other directions of the system, which is one of the limiting factors of the system bandwidth. The resonance frequencies and mode shapes of the 1-DOF FSM at other orders are obtained through modal analysis, as shown in Table 2 and Figure 7, respectively [19,20].

The first-order resonance of the 1-DOF FSM can be suppressed by controlling the motor, while the second-order and higher-order resonance are difficult to counteract through the control system. Therefore, the system’s working bandwidth is limited to within the second-order resonance frequency. To increase the system’s working bandwidth, as analyzed in Section 2, the first-order resonance frequency of the FSM should be reduced while increasing the second-order resonance frequency.

## 4. Optimization of Structure and Control

This section presents a method that combines the structural parameters of the FSM with controller parameters. Using this method, it is possible to achieve simultaneous first-order resonance suppression of the FSM based on the PID controller and natural frequency optimization of the FSM based on the NSGA-II algorithm.

### 4.1. PID Control for FSM

The object characteristics of the FSM obtained through system identification are as follows.
(11)$$G\left(s\right)=\frac{\omega_{n}^{2}}{s^{2}+2\xi \omega_{n}s+\omega_{n}^{2}}=\frac{K_{G}}{s^{2}+2\xi \omega_{n}s+\omega_{n}^{2}}.$$

To nullify the influence of the first-order resonance, PID control is employed to suppress the resonance. The PID controller is designed as follows.
(12)$$C_{PID}\left(s\right)=K_{p}+\frac{K_{i}}{s}+K_{d}s.$$

To enable the controller (12) to be physically implemented, an inertia element needs to be added, where $T_{c}$ represents half of the sampling frequency. The complete expression of the controller is as follows.
(13)$$C_{PID}\left(s\right)=\frac{K_{p}s+K_{i}+K_{d}s^{2}}{s(T_{c}s+1)}.$$

Combining (11)–(13), the open-loop transfer function and closed-loop transfer function of the system are shown below.
(14)$$F_{PID}\left(s\right)=C_{PID}\left(s\right)G\left(s\right)=\frac{K_{G}\left(K_{p}s+K_{i}+K_{d}s^{2}\right)}{s\left(T_{c}s+1\right)\left(s^{2}+2\xi \omega_{n}s+\omega_{n}^{2}\right)},$$
(15)$$T_{PID}\left(s\right)=\frac{K_{G}K_{d}s^{2}+K_{G}K_{p}s+K_{G}K_{i}}{T_{c}s^{4}+\left(2T_{c}\xi \omega_{n}+1\right)s^{3}+\left(T_{c}\omega_{n}^{2}+2\xi \omega_{n}+K_{G}K_{d}\right)s^{2}+\left(\omega_{n}^{2}+K_{G}K_{p}\right)s+K_{G}K_{i}}.$$

The characteristic equation of the closed-loop transfer function $L_{PID}\left(s\right)$ = 0 is as follows.
(16)$$L_{PID}\left(s\right)=T_{c}s^{4}+\left(2T_{c}\xi \omega_{n}+1\right)s^{3}+\left(T_{c}\omega_{n}^{2}+2\xi \omega_{n}+K_{G}K_{d}\right)s^{2}+\left(\omega_{n}^{2}+K_{G}K_{p}\right)s+K_{G}K_{i}=0.$$

The following equation discusses the stability of the system (16), thus providing the feasible set of controller parameters $k_{p}$, $k_{i}$, $k_{d}$, and $T_{c}$.
(17)$$\left\{\begin{array}{c} K_{p}>0,\\ K_{i}>0,\\ K_{d}>0,\\ T_{c}K_{p}-K_{d}<\frac{2\omega_{n}\xi }{K_{G}},\\ K_{i}<\frac{\left(2\omega_{n}\xi +K_{G}K_{d}-T_{c}K_{G}K_{p}\right)\left(\omega_{n}^{2}+K_{G}K_{p}\right)}{K_{G}}. \end{array}\right.$$

The proof of Equation (Equation 17) is provided in the Appendix A.

### 4.2. Multi-Objective Optimization Problem

To increase the working bandwidth of the 1-DOF FSM, it is essential to suppress the resonance through the controller and optimize the structure to reduce the first-order resonance frequency and increase the second-order resonance frequency. If both of these aspects can be achieved simultaneously, optimizing the system structure along with the controller parameters will greatly improve the efficiency of FSM research and development.

Since $f_{i}$ in (5) and $\omega_{n}$ in (11) are equivalent, they can be expressed as follows.
(18)$$\omega_{n}=f_{i}=\frac{1}{2\pi }\sqrt[{}]{\frac{K_{i}}{J_{i}}}.$$

By using (18), the controller parameters and the structure parameters can be combined to obtain a system function with an independent variable composed of the controller parameters $K_{p}$, $K_{i}$, $K_{d}$, $T_{c}$, and the structural parameters $J_{i}$, *R*, *t*, *b*, and *E*.
(19)$$F_{sys}=f\left(K_{p},K_{i},K_{d},Tc,J_{i},R,t,b\right).$$

$F_{sys}$ can be the system working bandwidth, as well as the system first-order resonance frequency or second-order resonance frequency. Therefore, we obtain a multi-objective optimization problem with the system working bandwidth, first-order resonance frequency, and second-order resonance frequency as the optimization objectives and the controller parameters and structural parameters as decision variables.

In reality, many optimization problems involve optimizing multiple objectives, and this type of problem is referred to as multi-objective optimization. The challenge of multi-objective optimization lies in optimizing multiple objectives simultaneously to obtain the best solution. Typically, this optimal solution is referred to as the Pareto optimal solution, which was introduced by Vilfredo Pareto in 1896 [21].

NSGA-II algorithm is applied as the solution to this problem. NSGA-II is a control-based multi-objective optimization algorithm proposed by Deb in 2002. It aims to solve the high computational complexity, lack of elitism strategy, and the need to specify sharing parameters in NSGA [22,23]. The NSGA-II algorithm utilizes fast, non-dominated sorting and crowding distance calculation methods, which can effectively perform multi-objective optimization with high efficiency. By selecting appropriate solutions from multiple non-dominated layers, NSGA-II can achieve very good multi-objective optimization results. The algorithm’s simple concept, easy implementation, and good adaptability make it an efficient choice for solving multi-objective optimization problems.

### 4.3. NSGA-II for FSM Controller and Structure Parameters

Based on the descriptions in Section 4.1 and Section 4.2, the optimization objectives are established as the first-order resonance frequency, the second-order resonance frequency, and the working bandwidth. Therefore, the following multi-objective optimization function is constructed.
(20)$$Object=\left\{\begin{array}{c} minf_{1}\left(x\right)=\omega_{n1},\\ maxf_{2}\left(x\right)=\omega_{n2},\\ minf_{3}\left(x\right)=1/\omega_{B}, \end{array}\right.$$
where $1/\omega_{B}$ is too small to be optimized during the iteration. Therefore, a coefficient ε is introduced to set the objective function value to the same order of magnitude, which means $minf_{3}\left(x\right)=\varepsilon /\omega_{B}$. Here we choose ε to be 10,000.

The decision variables are represented as follows.
(21)$$D=[K_{p},K_{i},K_{d},T_{c},J_{i},R,t,b,E].$$

The multi-objective optimization should be carried out under the premise of the structural stability of the FSM. The stability of the structure should satisfy maximum stress constraints and inequality constraints on the kinematic accuracy of flexible hinges.

#### 4.3.1. Maximum Stress Constraint

When the flexible hinge undergoes bending, the maximum stress in the hinge occurs at the minimum cut thickness *t* of the hinge, which has a stress concentration effect. According to the pure bending theory in material mechanics, the maximum stress in the flexible hinge is expressed as follows.
(22)$$\sigma_{max}=\frac{6K_{t}M_{max}}{t^{2}w},$$
where $K_{t}$ is the stress concentration factor, $M_{max}$ is the external torque applied to the hinge at the maximum deflection angle, and *w* is the width [24]. The calculation formula is as follows.
(23)$$\left\{\begin{array}{c} K_{t}={\left(1+\frac{at}{2b^{2}}\right)}^{\frac{9}{20}},\\ M_{max}=\frac{Ewt^{3}\cdot 10^{-3}}{4.316bp^{-0.4753}}. \end{array}\right.$$

Substituting (23) into (22) results in the following.
(24)$$\sigma_{max}={\left(1+\frac{at}{2b^{2}}\right)}^{\frac{9}{20}}\times \frac{6Et\cdot 10^{-3}}{4.316b{\left(\frac{a}{t}\right)}^{-0.4753}}.$$

For a circular flexible hinge, where *a* = *b* = *R*, Equation (Equation 24) is rewritten as follows.
(25)$$\sigma_{max}={\left(1+\frac{t}{2R}\right)}^{\frac{9}{20}}\times \frac{6Et\cdot 10^{-3}}{4.316R{\left(\frac{R}{t}\right)}^{-0.4753}}.$$

The material used for the 1-DOF FSM flexible hinge in this study is spring steel, with a yield strength in the range of 1570–1760 MPa. Therefore, taking $\sigma_{yield}=1600$ MPa and a safety factor $K_{safety}=3$. Substituting the yield strength $\sigma_{yield}$ (MPa) and safety factor $K_{safety}$, we obtain the following equations.
(26)$$c_{1}\left(x\right)=\sigma_{max}={\left(1+\frac{t}{2R}\right)}^{\frac{9}{20}}\times \frac{6Et\cdot 10^{-3}}{4.316R{\left(\frac{R}{t}\right)}^{-0.4753}}-\frac{1600}{3}\leq 0.$$

#### 4.3.2. Inequality Constraints on Kinematic Accuracy of Flexible Hinges

In the analysis of the kinematic accuracy of the flexible hinge, the displacement of the geometric center of the flexible hinge, that is, the displacement of the deflection center of the flexible hinge, is generally used as a quantitative measure of the kinematic accuracy of the flexible hinge [24]. In this FSM, the repeated positioning accuracy is $\delta_{p}\leq 8\mu$rad. The displacement of the center point, according to the formula of material mechanics, is obtained as follows.
(27)$$yc=\int \int \frac{M(x^{\prime })}{EI(x^{\prime })}dx^{\prime }dx^{\prime },$$
where
(28)$$\left\{\begin{array}{c} I\left(x^{\prime }\right)=\frac{wt^{3}\left(x^{\prime }\right)}{12},\\ M(x^{\prime })=M. \end{array}\right.$$

Equation (Equation 27) is transformed into a polar coordinate form as follows.
(29)$$\begin{array}{c} \frac{12Mb}{Ew}\int \int \frac{cos\varphi }{{\left(2a+t-2acos\varphi \right)}^{3}}d\varphi bcos\varphi d\varphi \\ =\frac{12Mb}{Ewt^{3}}\int_{-\frac{\pi }{2}}^{0}\int_{-\frac{\pi }{2}}^{\varphi }\frac{cos\beta }{{\left(2p+1-2pcos\beta \right)}^{3}}d\beta bcos\varphi d\varphi =\frac{6Mb^{2}}{Ewt^{3}}\cdot \frac{1}{1+2p}. \end{array}$$

The constraint expression for kinematic accuracy can be obtained from (28) as follows.
(30)$$c_{2}\left(x\right)=\frac{6\times 10^{-3}b}{4.316\cdot \left(1+\frac{2R}{t}\right){\left(\frac{R}{t}\right)}^{-0.4753}}-8\times 10^{-3}\leq 0.$$

Taking into account the system stability constraints required by (17), as well as the structural stability constraints required by (26) and (30), the decision variables in (21) need to satisfy the following conditions in total.
(31)$$\left\{\begin{array}{c} K_{p}>0,\\ K_{i}>0,\\ K_{d}>0,\\ T_{c}K_{p}-K_{d}<\frac{2\omega_{n}\xi }{K_{G}},\\ K_{i}<\frac{\left(2\omega_{n}\xi +K_{G}K_{d}-T_{c}K_{G}K_{p}\right)\left(\omega_{n}^{2}+K_{G}K_{p}\right)}{K_{G}},\\ c_{1}\left(x\right)=\sigma_{max}={\left(1+\frac{t}{2R}\right)}^{\frac{9}{20}}\times \frac{6Et\cdot 10^{-3}}{4.316R{\left(\frac{R}{t}\right)}^{-0.4753}}-\frac{\sigma_{yield}}{K_{safety}}\leq 0,\\ c_{2}\left(x\right)=\frac{6\times 10^{-3}b}{4.316\cdot \left(1+\frac{2R}{t}\right){\left(\frac{R}{t}\right)}^{-0.4753}}-8\times 10^{-3}\leq 0. \end{array}\right.$$

#### 4.3.3. Basic Idea of ISOC

To better understand the thought of this article, our basic idea is shown in Figure 8.

## 5. Simulation and Results

A PID controller (13) is designed based on the transfer function of the system (10), with $T_{c}=0.0001$; the formula is expressed as follows.
(32)$$C_{PID}\left(s\right)=\frac{K_{p}s+K_{i}+K_{d}s^{2}}{s(0.0001s+1)}.$$

The combination of Equations (10) and (32) results in the open-loop transfer function of the system as follows.
(33)$$F_{PID}\left(s\right)=C_{PID}\left(s\right)G\left(s\right)=\frac{8.463\times 10^{4}\times \left(K_{p}s+K_{i}+K_{d}s^{2}\right)}{s\left(0.0001s+1\right)\left(s^{2}+53.12s+7.818\times 10^{4}\right)}.$$

In order to suppress the first-order resonance of the system, the parameters of the controller need to be adjusted. The values of $K_{p}$, $K_{i}$, and $K_{d}$ obtained through automatic tuning of the PID are as follows.
(34)$$K_{p}=7.5344,K_{i}=847.6144,K_{d}=0.016743.$$

Under these parameter settings, the system has a phase margin of $75^{\circ }$, ensuring stability and effectively eliminating the first-order resonance. The characteristics of the system and the closed-loop transfer function after tuning are displayed in Figure 9.

The structural parameters of the flexible hinge are as follows (units: mm).
(35)R=7.5,t=0.5,b=14.

The moment of inertia corresponding to the first-order resonance is $J_{1}=169\times 10^{-6}$ kgmm^2^, and the moment of inertia corresponding to the second-order resonance is $J_{2}=4400\times 10^{-6}$ kgmm^2^. Due to the relatively minor impact of changes in the flexible hinge structure on the moments of inertia in different rotational directions, we assume $J_{1}$ and $J_{2}$ to be constant values. Additionally, the material used for the flexible hinge is spring steel, with Young’s modulus of $E=2.1\times 10^{11}$ Pa.

The initial values of the system decision variables have been determined through (34) and (35), and it is necessary to establish the upper and lower bounds for these decision variables. Due to the interdependent nature of the parameters in constraint (31), it is not possible to obtain the range of each parameter individually. Therefore, based on (34) and (35), the upper and lower bounds of the decision variables are expanded, and an algorithm is employed to eliminate individuals that do not satisfy the constraints in Equation (Equation 31), thus meeting the requirements. As a result, the upper and lower bounds of the decision variables are determined as follows: (36)$$\left\{\begin{array}{c} l_{limit}=\left[4.5,700,0.01,0.005,0.00045,0.01\right],\\ u_{limit}=\left[10,1000,0.02,0.01,0.001,0.02\right]. \end{array}\right.$$

The programming and execution were carried out using Matlab (Version R2022b), ultimately generating the multi-objective optimization results, as shown in Figure 10.

A group of individuals with a relatively small first-order resonance frequency and a large interval between first-order resonance frequency and second-order resonance frequency are selected from the solution sets as the optimal result. The corresponding structural parameters and controller parameters are as follows:(37)$$K_{p}=8.7834,K_{i}=886.8503,K_{d}=0.01,R=5,t=0.45,b=20.$$

The structural parameters and control parameters corresponding to the structure before and after optimization are shown in Table 3.

It can be seen that the interval between the first-order resonance frequency and the second-order resonance frequency of the system before optimization is 122 Hz. Butthrough optimization, the interval between the first-order resonance frequency and the second-order resonance frequency of the system is 260 Hz, which increases by 113% on the basis of the initial structure and greatly expands the system bandwidth.

To validate the effectiveness of the optimization algorithm, we conducted verification from two aspects. Firstly, we constructed the corresponding 3D model of the optimized object based on the structural parameters in Table 3 and obtained the modal analysis results of the optimized model through ADAMS/Vibration, as displayed in Table 4. The first-order resonance frequency of the optimized model is 52.393 Hz, with an error of 0.7% compared to the optimization result, and the second-order resonance frequency is 321.57 Hz, with an error of 3.06% compared to the optimization result.

Secondly, the optimized model was co-simulated to obtain its open-loop frequency response characteristics. The optimized PID parameters in Table 3 were used to adjust the PID controller parameters, and the closed-loop system under the new structure was obtained. Figure 11 shows the characteristics of the FSM and the closed-loop transfer function. It is evident that the first-order resonance of the optimized system is greatly suppressed, indicating that the algorithm is also effective for the optimization of controller parameters.

## 6. Conclusions

To ameliorate the research and development efficiency of the FSM, an integrated optimization of structure and control (IOSC) methodis proposed. This allows for the integratedoptimization of FSM structural parameters and controller parameters. Using this approach, the first-order resonance of the FSM is suppressed by a PID controller, and the NSGA-II algorithm is utilized to optimize the first-order resonance frequency, second-order resonance frequency, and closed-loop bandwidth of the FSM as optimization objectives. This enables the simultaneous suppression of the first-order resonance of the FSM while optimizing the system’s bandwidth.

The group with the maximum interval between the first and second-order resonance frequencies is selected as the optimization result. In this work, the interval increases by 113% on the basis of the initial structure, which is a great improvement and greatly expands the system bandwidth. Based on the controller parameters and structural parameters in the optimization result, remodeling and simulation are performed, and the simulation results show very small errors compared to the optimization results. Hence, we believe that the method presented in this study provides an efficient optimization approach for the development of FSM.

## Figures and Tables

**Figure 1 micromachines-15-00298-f001:**
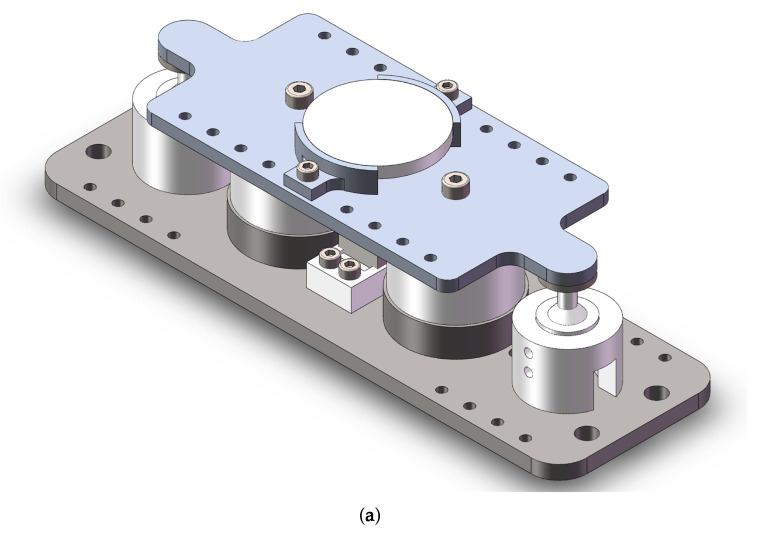
Structure of a 1-DOF FSM. (**a**) The 3D model of the 1-DOF FSM. (**b**) Exploded view of the FSM.

**Figure 2 micromachines-15-00298-f002:**
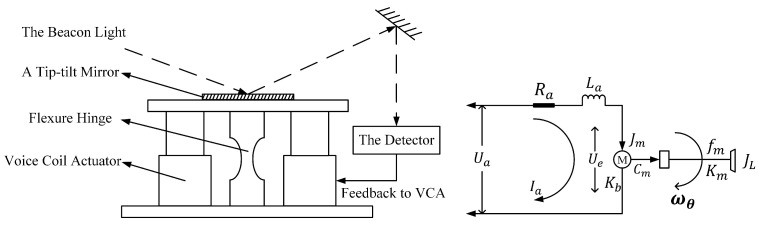
Structural principles of a 1-DOF FSM and physical model structure of the controlled device.

**Figure 3 micromachines-15-00298-f003:**
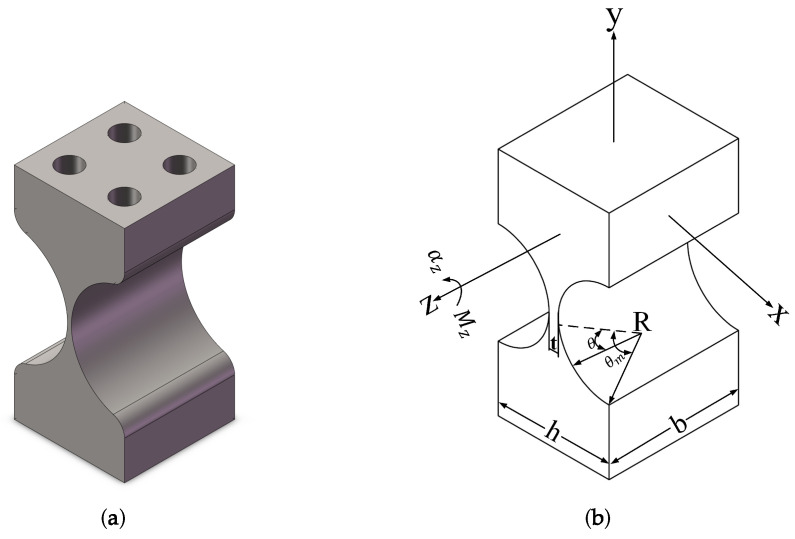
(**a**) 1-DOF tangential circular flexible hinge. (**b**) Force and deformation of the flexible hinge.

**Figure 4 micromachines-15-00298-f004:**
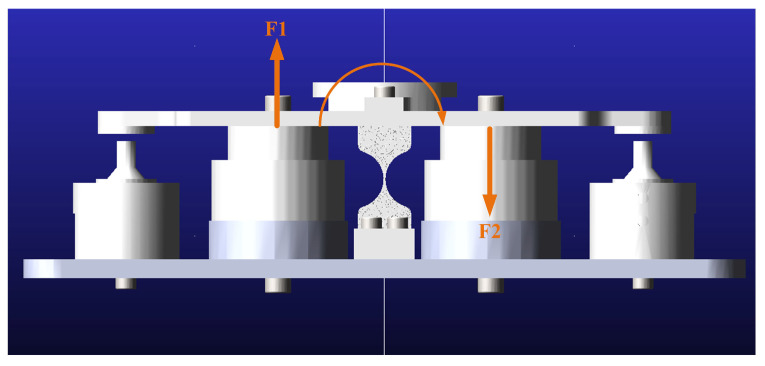
The FSM dynamic model in ADAMS.

**Figure 5 micromachines-15-00298-f005:**
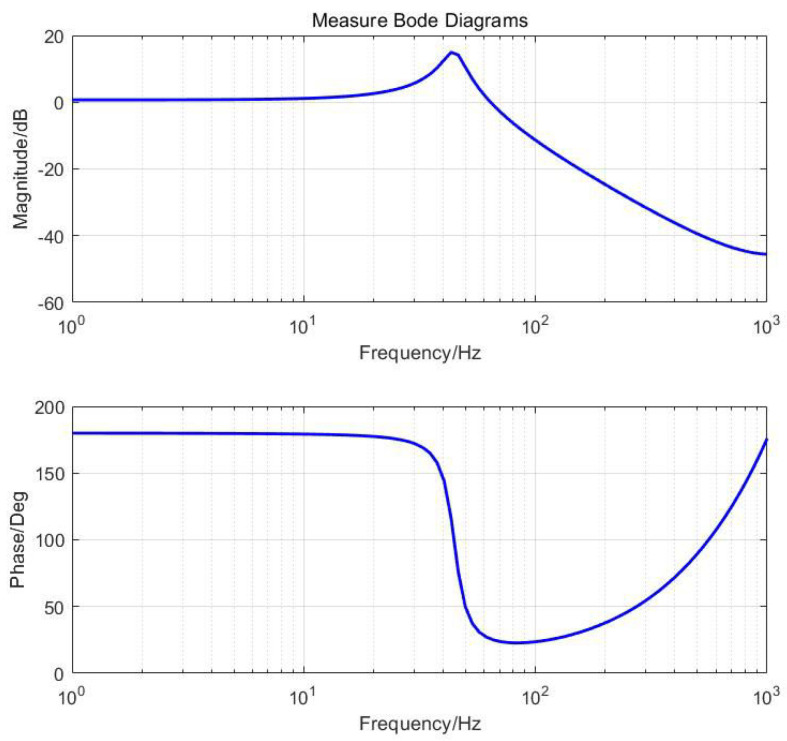
The open–loop frequency response of the FSM around Z-axis rotation.

**Figure 6 micromachines-15-00298-f006:**
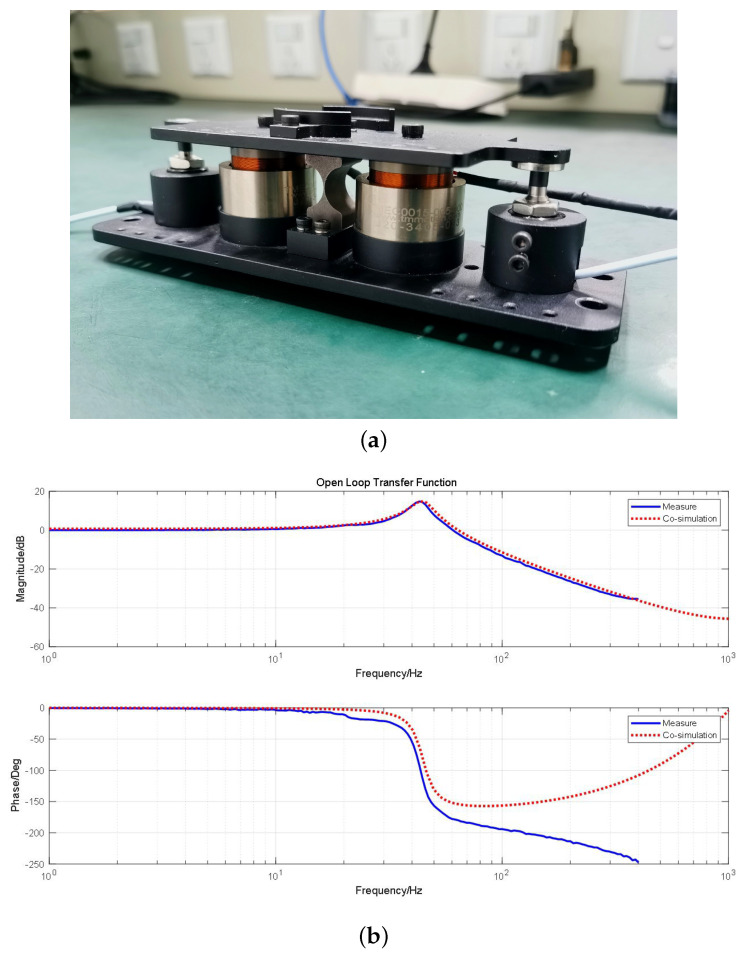
(**a**) A physical protoype of the FSM. (**b**) Comparison between co–simulation and measurement.

**Figure 7 micromachines-15-00298-f007:**
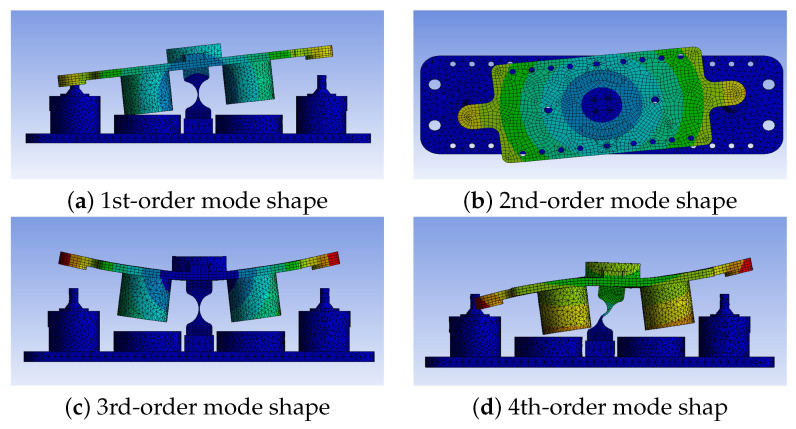
The mode shapes of the 1-DOF FSM for the first four orders.

**Figure 8 micromachines-15-00298-f008:**
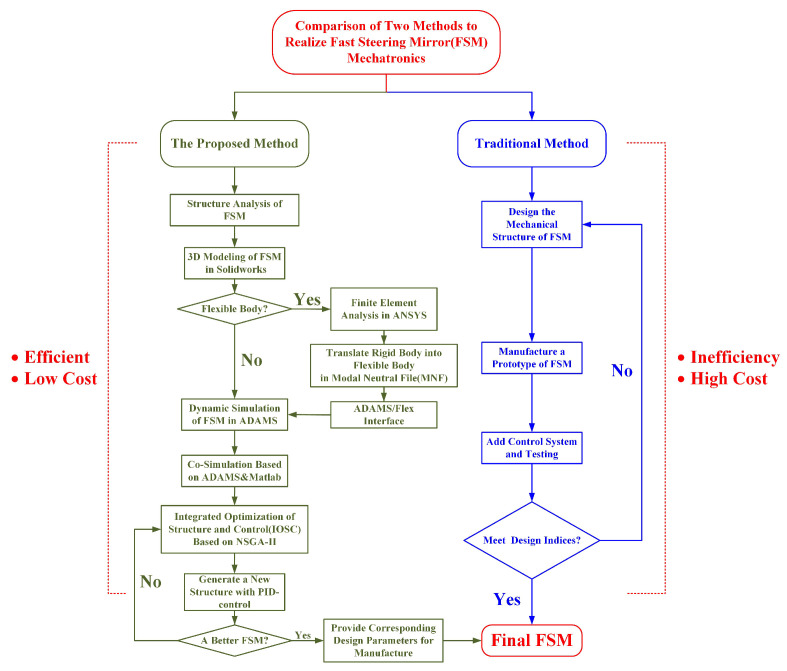
Basic idea of this article.

**Figure 9 micromachines-15-00298-f009:**
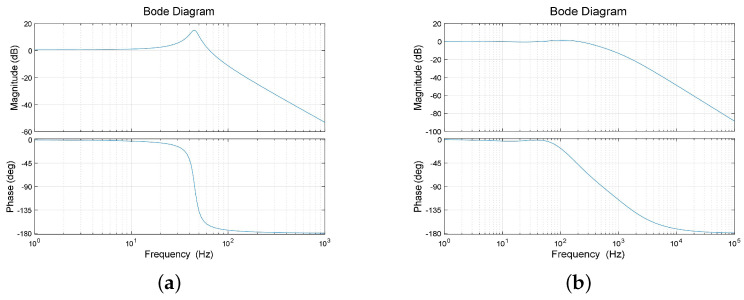
(**a**) The system open–loop characteristic before optimization. (**b**) The system closed–loop characteristic before optimization.

**Figure 10 micromachines-15-00298-f010:**
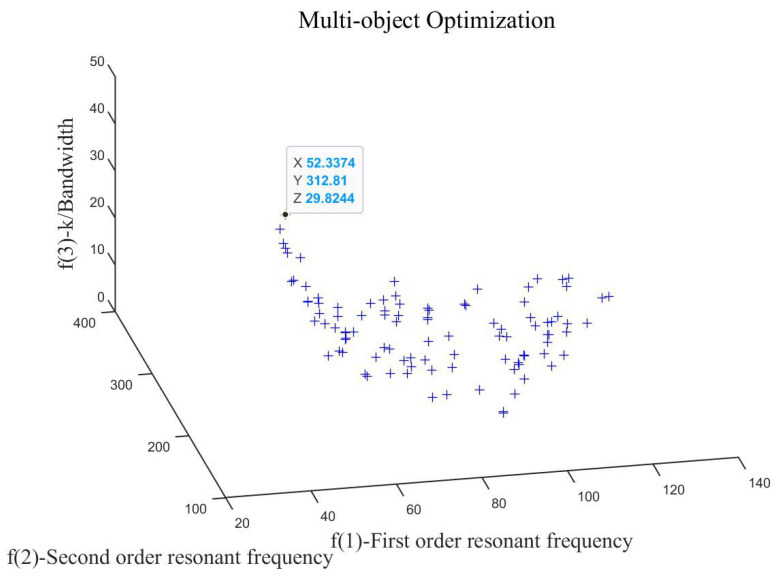
The results of multi-objective optimization.

**Figure 11 micromachines-15-00298-f011:**
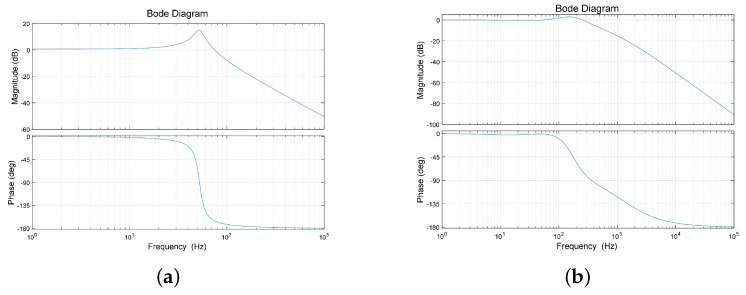
(**a**) The system open–loop characteristic after optimization. (**b**) The system closed–loop characteristic after optimization.

**Table 1 micromachines-15-00298-t001:** The 1-DOF FSM Simulated Performance.

Property	Simulated Performance
FSM Size (Length, Width, Height)	190 mm, 60 mm, 60 mm
Mirror Diameter	38 mm
Angular Range	±3.9 mrad
Repeated Positioning Accuracy	8 μrad
Step Response Time	0.36 s

**Table 2 micromachines-15-00298-t002:** Resonance frequencies of different orders of the initial structure.

Mode Number	Resonance Frequency (Hz)
1	44.925
2	166.984
3	1228.71
4	1498.75

**Table 3 micromachines-15-00298-t003:** Dynamic performance of the system before and after optimization.

	R	t	b	$K_{p}$	$K_{i}$	$K_{d}$	1st-Order Resonance Frequency	2nd-Order Resonance Frequency
Initial FSM	7.5	0.5	14	7.53	847.6	0.0167	45	167
Optimized FSM	5	0.45	20	8.78	886.8	0.01	52	312

**Table 4 micromachines-15-00298-t004:** Resonance frequencies of different orders after optimization.

Mode Number	Resonance Frequency (Hz)
1	52.393
2	321.923
3	2111.67
4	2108.86

## Data Availability

Data is contained within the article.

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
