# Peer review of "Integrated Optimization of Structure and Control for Fast Steering Mirrors"

_micromachines, 2024, doi:10.3390/mi15030298_

Round 1
Reviewer 1 Report
Comments and Suggestions for Authors
This paper presents a method to achieve simultaneous optimization of both mechanical structure and control system on the basis of a fast steering mirror. The proposed integrated can provide better time and cost effectiveness compared with traditional method.
The review comments are summarized as follows.
1. What is the reason chosen as 1-DOF FSM system? Can the 1-DOF FSM be extend to higher DOF FSM to meet universal application asserted by the authors? This issue should be clarified.
2. The legends of each part in Figure 1 are required.
3. The more details are required for the FSM including the size, controllable angle, response time, etc.
4. In fact, the system shown in Figure 1 is nonlinear. Why did the authors consider this system as a linear model? Some assumptions are required to use as a linear model.
5. How to get the numerical values of Eq. (10)? Are the values obtained from measurement or simulation (including FEM)?
6. A flow chart showing the structural optimization and control is shown in Figure 8. The presentation of a comparative flow chart between the proposed and traditional method will be much better to understand the main difference and some advantages of the proposed method.
7. How to choose the control gains of PID controller and other numerical values?
8. Figure 10 is not clear. It should be redrawn to be more clearly including the legend.
9. How to choose the values of the initial FSM in Table 2? The effectiveness of the optimal FSM heavily depends on the initial values.
10. The authors asserted in Introduction that the proposed method is more advanced in the sense of the cost effectiveness and simultaneous prediction of the control performance of the FSM under different structure compared with the previous studies [15, 16] which are featured by a serial design approach. However, the authors did not validate the asserted benefits of the proposed method compared with the traditional method. Thus, it is very difficult to figure out the main technical contribution of this work. This reviewer strongly recommends the authors to clearly present the advantages of the proposed method in terms of quantitative values instead of a rough statement.
Comments on the Quality of English LanguageEntensive iimplrovement is required.
Author Response
请参阅附件。

Reviewer 2 Report
Comments and Suggestions for Authors
This paper proposes an integrated optimization of structure and control method. In this paper, the structure parameters and control parameters of the fast steering mirrors are combined together to optimize, which efficiently accelerate the development of the fast steering mirrors. Some suggestions are listed as follows:
1. There are inappropriate expressions in the text, which requires a review. For example, on page 4, “When there is no motor in the system” should be used instead of “When the system does not have a motor”.
2. Please enlarge the notations of Figs. 3, 6, 7 and 10, to improve readability.
3. The phase frequency characteristics in Fig. 5 do not match the usual phase frequency characteristics of the second-order oscillation system described in equation (4), it is suggested to add some explanations.
4. The stability of the system needs to meet the conditions in equation (31), can you please explain how the optimized parameters meet these conditions?
5. How does this paper select the PID controller parameters in equation (34) as the initial values?
Comments on the Quality of English Languageno
Round 2
Reviewer 1 Report
Comments and Suggestions for Authors
This paper has been well revised and hence it is acceptable in its current form.
Comments on the Quality of English LanguageEnglish editing is required in the sense of grammatical errors.